# Using Trait-Based Approaches to Assess the Response of Epedaphic Collembola to Organic Matter Management Practices: A Case Study in a Rubber Plantation in South-Eastern Côte d’Ivoire

**DOI:** 10.3390/insects13100892

**Published:** 2022-09-30

**Authors:** Aymard Kouakou Kouakou, Jérôme Cortet, Yeo Kolo, Alain Brauman

**Affiliations:** 1Station D’écologie de Lamto, Université Nangui Abrogoua, Abidjan 02 BP 801, Côte d’Ivoire; 2Eco & Sols, University Montpellier, CIRAD, INRA, IRD, Montpellier SupAgro, F-34398 Montperliier, France; 3UMR CEFE 5175, University of Montpellier, EPHE, University Paul-Valéry Montpellier, Route de Mende, CEDEX, F-34199 Montpellier, France

**Keywords:** collembola, morphotypes, bioindicators, rubber plantation, functional trait, organic matter

## Abstract

**Simple Summary:**

While studies based on the taxonomic facet of biodiversity have already proven their value in understanding soil functioning, studies focusing on the functional facet based on the traits of organisms are scarce in the Ivorian context. Among soil organisms, springtails play an important role in soil functioning and are a useful bioindicator for assessing the impact of land use change and agricultural practices on soil biodiversity. However, their taxonomy is very poorly known in Côte d’Ivoire. The functional trait approach is therefore a relevant alternative for assessing the response of springtails communities to organic matter management in tree plantations. The aim of this study was to determine how different input of organic matter in the form of logging residues and legumes influence the body size and functional diversity of springtails. Our results showed a high functional richness and body size of springtails in the practice with trunks and large branches (R2L1). Functional traits are useful to assess the effects of agricultural practices on springtails communities.

**Abstract:**

We used trait-based approaches to reveal the functional responses of springtails communities to organic matter inputs in a rubber plantation in Côte d’Ivoire. Pitfall traps were used to sample springtails in each practice. The results showed that the total abundance of springtails increased significantly with the amount of organic matter (R0L0 < R2L1). Larger springtails (body length, furca and antennae) were observed in plots with high organic matter. Practices with logging residues and legume recorded the highest functional richness. The principal coordinate analysis showed different functional composition patterns between practices with logging residues (R1L1 and R2L1) and those without inputs (R0L0 and R0L1). This difference in functional composition (PERMANOVA analysis) was related to the effect of practices. These results highlight the pertinence of the functional trait approach in the characterization of springtail communities, a bioindicator of soil health, for organic matter management practice.

## 1. Introduction

Land-use change is the cause of biodiversity loss and the depletion of terrestrial ecosystems [1]. The effects of these changes are mostly driven by the conversion of natural ecosystems to agricultural lands. Indeed, the conversion of forests into agricultural land leads to a degradation of the physical and chemical quality of soils [2]. In West Africa, particularly in Côte d’Ivoire, land-use changes lead to a decrease of total organic carbon and soil pH [3], of the abundance, richness and diversity of litter and soil arthropods [4,5]. Furthermore, the conversion of primary or secondary forests to rubber plantations appears to reduce the diversity and density of soil macroinvertebrates [6,7,8].

Successive rubber plantation cycles of 25 to 40 years [9], lead to decline of soil functions [10] and a continuous loss of soil biodiversity [11]. One of the ways to mitigate soil degradation related to intensive land use is to add organic matter. Indeed, the addition of organic matter such as logging residues and legumes after a rubber plantation cycle promotes the resilience of the main soils functions (carbon transformation, soil structure maintenance and nutrient cycling [12]).

Soil biodiversity plays a crucial role in maintaining soil functioning and ecosystem services [13]. The return of organic matter to the soil provides a suitable microclimate, provides new ecological niches and resources to maintain this biodiversity [14]. However, the effects of organic matter restitution on soil biodiversity in tree plantations are still unclear. Thus, understanding the impacts of organic matter input on soil biodiversity can contribute to the development of a sustainable management program in tree plantations.

Among soil organisms, springtails are key bioindicators of soil functions [15] such as nutrient cycling [16,17] and organic matter decomposition [18]. Springtails are also considered as relevant bioindicators of land use change [19,20,21]. They are also used as biological indicators to assess the impact of different agricultural practices on soil biodiversity [22,23,24,25].

Africa is an immense reservoir of biodiversity, representing around a quarter of the world’s biodiversity [26] due to its varied ecosystems (tropical forests, savannahs, deserts, mangroves and mountain grasslands). However, the problem of identification arises for many taxa that are still poorly known. Ecological research faces serious difficulties in identifying species due to the lack of identification keys for many species [27]. In Côte d’Ivoire, research has been conducted in this context to identify arthropod species on the basis of their taxonomy [28,29,30,31]. Concerning springtails communities, despite recent works by Zon et al. [32,33], their taxonomy is very poorly established in Côte d’Ivoire.

The functional traits refer to the characteristics of a species that can influence the performance of an individual, i.e., its growth, reproduction or survival [34]. These traits govern the responses of individuals to disturbances in their environment [34,35]. Approaches based on the use of functional traits can improve the understanding of the responses of soil invertebrates to environmental disturbances and provide complementary information to that provided by taxonomic approaches [20,36,37]. Functional traits also provide a better understanding of how abiotic factors determine the species assemblage of a soil community and predict how it will develop in response to changes in habitat [38].

Therefore, in the Côte d’Ivoire context, where the taxonomy is poorly known, the functional traits of springtails could be used as indicators of agricultural practices, particularly those linked to the management of organic matter, which is the basis of food webs, probably leading to functional community modifications.

The objective of this study was to determine how the input of organic matter, such as logging residues and legumes, modulates the functional responses of springtails. We hypothesized that (1) the massive presence of carbon would induce a greater abundance of springtails, as a result of increasing the amount of carbon and energy available for the soil food webs; (2) the high presence of organic matter on the surface is a preferential habitat for large individuals with a well-developed furca; (3) the presence of organic matter on the surface would increase the number of ecological niches and thus promote a higher functional richness and diversity. We conducted this study 12 months after the addition of logging residues and legumes in a large-scale field experiment in a rubber tree plantation in Côte d’Ivoire.

## 2. Materials and Methods

### 2.1. Study Site

This study was conducted in the South-East of Côte d’Ivoire on a rubber plantation. This plantation belongs to the “Société Africaine de Plantations d’Hévéas” (SAPH). The site is located (latitude 5°31′02.5″ N, longitude 3°29′46.2″ W) in the department of Grand Bassam with an area of 5503 ha. The climate of the region is sub-equatorial, adapted to rubber plantation, with four seasons. There is a large rainy season from April to June; a small dry season from July to August; a small rainy season from September to October and a large dry season from November to March. Annual rainfall varies between 1700 mm and 1900 mm and the annual temperature varies between 24 and 27 °C. The soil has a sandy-silty texture with 11% clay and an acidic pH between 4 and 5 at 0–10 cm depth [12]. The topography is homogeneous with low slopes (<5%).

### 2.2. Experimental Design

We set up an experimental design after the clearcutting of the previous rubber plantation (40 years old) using bulldozers in November 2017. In this site, the natural rainforest was the previous land-use type, and the logged stand consisted of the first cycle of rubber trees. The experimental design has already been described by Perron et al. [12]. However, in this paper, only one site (SAPH) was considered. The experimental design includes 4 practices replicated in 4 random blocks, resulting in 16 plots as follows:−R0L0: all rubber residues (R) removed from the plot, no legume (L);−R0L1: all rubber residues removed from the plot, legume (*Pueraria phaseoloides*);−R1L1: stumps, fine branches (less than 20 cm diameter) and leaves from the previous plantation left in the inter-row, legume (*Pueraria phaseoloides*);−R2L1: no rubber tree residues removed (leaves, trunk and stumps left), legume (*Pueraria phaseoloides*).

In the experimental setup described, only 3 blocks were studied in our case.

The practices defined in the experimental setup constitute an organic matter gradient. This gradient is related to the gradual increase in the amount of organic matter added between practices from R0L0 (no residues) to R2L1 (practice with the highest amount of residue) as well as to a gradual evolution of the quality in terms of C/N between R0L1, a practice with legume only (*Pueraria phaseoloides*) with a low C/N, and R2L1, which contains woods (trunks and branches) with a high C/N.

Rubber tree residues were put in the inter-rows. Before the tree logging, an inventory of living trees was realized. In «R1L1» and «R2L1» practices, logging residues were set up according to the results of this inventory in order to have a similar quantity of residues (leaves, trunk and stumps left) per practice in the 4 blocks and then guarantee the homogeneity of the experimental design. The number of trees per plot was 30 and 28, respectively, in R1L1 and R2L1 practice. The C stock in the practice with rubber residues was 36 t ha^−1^ in R1L1 and 97 t ha^−1^ in R2L1. The legume (*Pueraria phaseoloides*) was broadcast (10 kg ha^−1^ of wet seed) in February 2018. 

### 2.3. Sample Collection

Sampling was carried out in November 2018 (12 months after the practices were set up), when the rubber tree residues on the ground were fully decomposed and the legume well developed. Pitfall traps were chosen for the capture of springtails in each practice. This method is efficient for sampling the surface active springtails as already shown by different authors [39,40]. In this study, plastic cups of 3.5 cm diameter were used as pitfall trap. The volume of each pitfall trap was 300 mL and each one was filled to about 1/3 of its volume with 70% ethanol. The traps were left in activity for two days. Given the climatic constraints (intense rainfall) prevailing on this site, we shortened the activity times of pitfall to two days to avoid flooding with sands and water in the traps. However, usually, the activity time of the pitfall trap is longer [39,40]. A sampling plot (25 m × 10 m) was defined to limit border effects in each practice. In each sampling plot, six pitfall traps were placed in two lines of three traps, only in the windrows for sampling epedaphic or atmobiotic springtails [41]. In each line, the pitfall trap was spaced to 10 m and the distance between the lines was 8 m (Figure 1). A total of 6 samples were taken in each plot.

### 2.4. Measurement of the Springtails Traits

Springtails were characterized by morphotypes rather than species for functional trait measurements. The functional traits of springtails were measured directly on individuals collected in the field. We selected traits related to dispersal ability, life form and habitat preference of springtails [20,42,43]. The selected traits are summarized in Table 1. We used observed trait data to calculate the functional diversity (FD) indices proposed by Villéger et al. [44]. The functional diversity indices calculated in this paper are as follows: −The functional richness (FRic) corresponds to the volume of functional space occupied by species (abundance is not involved);−The functional divergence (FDiv) corresponds to a degree of niche differentiation among species within communities;−The functional evenness (FEve) measures the regularity of the distribution of abundance in functional space;−The functional dispersion (FDis) is a “pure” estimator of the dispersion in trait combination abundances.

### 2.5. Statistical Analysis

Abundance data were transformed to Log (x + 1) for normal distribution and homogeneity of variances. One-way analysis of variance (ANOVA) and comparisons of measured traits and functional diversity indices were performed using Tukey’s test (*p* < 0.05) with the agricolae package [45]. All variables were tested for normality and homogeneity of the variance of the data using Shapiro–Wilk and Levene tests, respectively. Quantitative measurements of body length, antenna length and furca length (in µm in Table 1) were used to characterize the size of the springtails. We used the dbFD function of the FD package [46] to compute functional diversity (FD). dbFD uses principal coordinates analysis (PCoA) to return PCoA axes, which are used as ‘traits’ to compute FD. dBFD computes FD indices, including the three indices of Villéger et al. [44]: functional richness (FRic), functional evenness (FEve) and functional divergence (FDiv). It also computes functional dispersion (FDis) [46] and the community-level weighted means of trait values (CWM), an index of functional composition. Following this computation, the results are presented in the form of one weighted sample for each practice per block. The CWM was calculated from the observed qualitative traits (Body modification, Dens, Mucro, Ocelli, Post Antennal Organ, Pigmentation, Scales, Empodial appendage) and their different attributes. The calculation of the community-weighted mean trait attribute values was based on two matrices (practice by morphotype and morphotype by trait). The CWM were used in a principal coordinate analysis (PCoA) to explore the functional composition patterns of springtails according to practice types [47]. This analysis was based on a Euclidean distance matrix. Then, a non-parametric multivariate analysis of variance (PERMANOVA) using the vegan package [48] was performed to test for differences in functional composition of springtails. All statistical analyses were performed with RStudio software, version 1.3.1093 [49].

## 3. Results

### 3.1. Variation of Total Springtails Abundance with Amount of Organic Matter

The total abundance of springtails varied significantly (ANOVA, F = 3.35; Df = 3; *p* = 0.02) between the different practices (Figure 2). The abundance of springtails observed in the R2L1 practice was four times greater compared to the one observed in the practice without organic matter (R0L0). R0L1 and R1L1 presented intermediate abundance.

### 3.2. Springtails Size Response to Logging Residue and Legume Input

The measured traits show that the size of the springtails increases with the amount of organic matter (Figure 3). A significant difference in body length (ANOVA, F = 13.41; Df = 3; *p* = 3.52 × 10^−8^), antenna length (ANOVA, F = 9.54; Df = 3; *p* = 5.79 × 10^−6^) and furca length (ANOVA, F = 10.41; Df = 3; *p* = 2.11 × 10^−6^) of springtails was observed between the different practices (Figure 3A–C). The largest springtails sizes were observed in the R2L1 practice, intermediate in the R1L1 and R0L1 practices and smallest in the R0L0 practice (without residues).

### 3.3. Response of the Functional Diversity Indices of Springtails to Logging Residues and Legumes

Functional diversity indices varied significantly between practices (Figure 4). Significant differences (ANOVA, F = 5.47; Df = 4; *p* = 0.02) in functional richness (FRic) were observed between practices with the highest values in practices with rubber residues (R1L1 and R2L1, Figure 4A). Significant difference in functional divergence (FDiv, ANOVA, F = 28.60; Df = 4; *p* = 1.98 × 10^−4^) was observed between practices with residues and/ or legumes (R0L1, R1L1 and R2L1) and those without (R0L0, Figure 4B). A significant difference in functional evenness (FEve, ANOVA, F = 4.13; Df = 4; *p* = 0.04) was also observed between practices with legumes only (R0L1) and residues (R2L1, Figure 4C). Functional dispersion (FDis) was significantly different (ANOVA, F = 41.16; Df = 4; *p* = 5.97 × 10^−5^) between practices with rubber residues (R2L1) and those without (R0L0, Figure 4D).

### 3.4. Response of the Functional Composition (CWM) of Springtails to the Organic Matter Gradient

The community weighted mean trait values (CWM) were grouped into an ordination diagram by applying a principal coordinate analysis (PCoA). The practices plotted in the Principal Coordinate Analysis represent plots where springtails were sampled. According to the PCoA analysis, the first axis explains 91.23% and the second axis explains 8.76% of the total variance. The functional composition patterns are separated by the absence (R0L0 and R0L1) and amount of rubber residues (R2L1, R1L1, Figure 5). The functional composition shows a significant difference between the practices (PERMANOVA, F = 1.95, R² = 0.69, *p* = 0.001).

## 4. Discussion

This study evaluated the functional response of springtails communities to organic matter inputs such as logging residues and legumes after a rubber plantation cycle. The results indicate that the abundance, diversity and functional composition of springtails varied significantly with the amount of organic matter input.

The large abundance of springtails in the practice with rubber tree residues (R2L1) is believed to be the result of the high amount of organic matter in this plot. The large amount of OM can induce a microclimate favorable to many springtails. These favorable conditions would allow a very high colonization rate with a rapid increase in springtail populations [50]. Rubber tree residues (trunk, leaves, stump and fine branch) and the legume on the ground constitute large quantities of accumulated OM that can harbor a greater abundance of springtails [51]. In addition, this high OM presence may increase the resource heterogeneity necessary for increased springtail abundance [52]. In the R0L0 practice, the absence of OM leads to a degradation of the habitat quality, which prevents many springtails to colonize this habitat [53]. Indeed, in the absence of vegetation cover and OM, the soil is subject to high solar radiation and temperature, which can impact on the abundance of springtails [54].

The body size of springtails increased significantly with the amount of OM. Such size differences could reflect a high abundance of epedaphic springtails in the presence of a large amount of OM on the soil surface [55]. Our results are in accordance with the studies of Yu et al. [55], who showed that manure input affects the functional composition of the community, favoring more active and mobile species of springtails characterized by a well-developed furca. Our results suggest that after a cycle of rubber plantations, organic matter management practices can help increase the abundance of springtails [56]. The results also suggest that the quality and availability of resources, rather than their quantity, control springtail communities. Furthermore, the sampling method could have an influence on our results as pitfall traps tend to capture epedaphic and more mobile springtails. Trends might be different with other non-selective methods (soil sampling) or a combination of methods [39,57]. 

The amount of OM would probably be the main factor that determines springtails communities since we observed a significant difference in functional diversity indices between practices in our study. The high functional richness (FRic) in practices R1L1 and R2L1 can be explained by the creation of new ecological niches in the presence of OM. These results would reflect that springtails occupy different amounts of niche space according to the practices. The functional divergence (FDiv) being different according to the practices seems to indicate that there are degrees of niche differentiation, and consequently, competition for the available resources. Indeed, the difference in body size of springtails in relation to the presence of OM can influence both the distribution of resources and trophic status within the food web [58]. The significant variation in functional evenness (FEve) would result from the difference in abundance of springtails between practices. This difference could influence the degree of distribution of springtails communities according to ecological niches to allow efficient use of the full range of available resources [59]. The significantly higher functional dispersion (FDis) in practices with residues than those without reflects a proportion of epedaphic springtails with the most extreme trophic niches in the community [41,60]. This variation in FDis would also be the result of variation in microclimate conditions in the practices, which influence species abundance. Some abiotic factors or changes in soil structure that we did not measure in this study could also have an influence on our results, as shown by Susanti et al. [61].

The functional composition of springtails communities was not significantly different in the two practices without rubber residues (R0L0 and R0L1). These results suggest that the springtails communities in these practices are functionally similar. Thus, the difference in practices between R0L0 and R0L1 is not sufficient to induce a significant functional change of the communities. An additional explanation for the non-differentiation in the functional composition of springtails in these two practices refers rather to the effects of adaptation and the range of ecological niches of the species, which may in fact decouple their distribution from environmental constraints [62]. The different functional composition in the practices with residues (R1L1 and R2L1) can be explained by the fact that the quantity and quality of OM, being both food and habitat for soil fauna, modulate this composition and the structure of the soil food web [63,64]. Differences in functional composition between practices would be associated with changes in trophic niches in springtails communities [41,60]. The differentiation of the trophic niches of springtails according to the practices could be explained by the regulatory power of the latter in relation to the microbial communities responsible for the decomposition of OM [65]. Indeed, depending on the amount of OM, the microbial biomass may differ from one practice to another and induce different springtail populations [66].

## 5. Conclusions

This study assessed the functional response of springtail communities to organic matter input after a 40-year rubber plantation cycle. Among the different practices, the one with trunks, branches and leaves, combined with the legume (R2L1) showed the greatest abundance of springtails. The results showed that in the presence of a large amount of organic matter, large springtails with a well-developed furca colonized the habitat. The presence of rubber tree residues (trunks, fine branches, leaves and stumps) induced a higher functional richness. The functional composition patterns showed that the springtail communities occupied different ecological niches. Functional trait analysis is a good alternative for studying springtails distribution and soil health. The study highlights the pertinence of the functional approach in the characterization of a community whose taxonomy is unknown.

## Figures and Tables

**Figure 1 insects-13-00892-f001:**
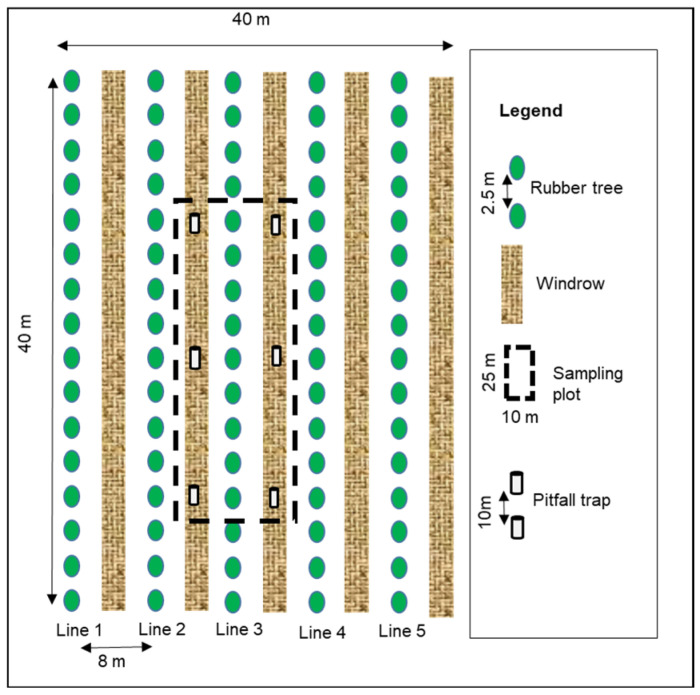
Sampling design in each plot.

**Figure 2 insects-13-00892-f002:**
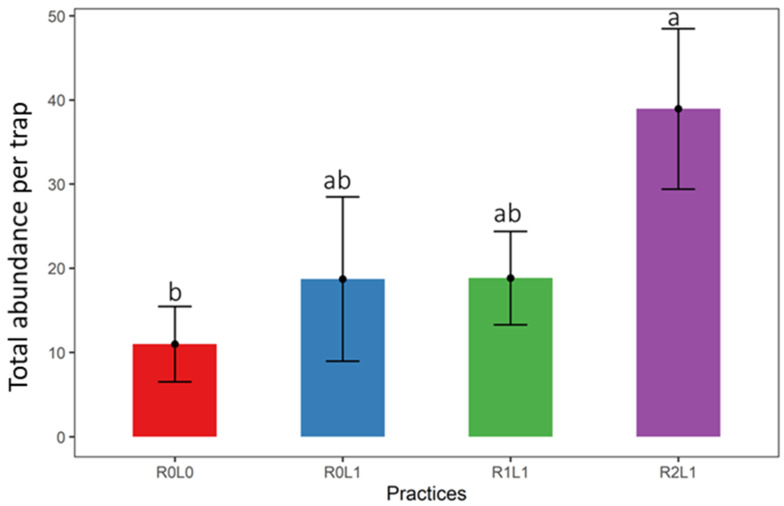
Variation in total abundance of springtails between practices. Vertical lines represent standard errors for each practice (*n* = 18). Different letters indicate significant differences according to the Tukey test. R0L0—no residue or legume in the plot, R0L1—Legume (*Pueraria phaseoloides*) only, R1L1—Legume + stump + leaf + fine branches, R2L1–R1L1+ trunk.

**Figure 3 insects-13-00892-f003:**
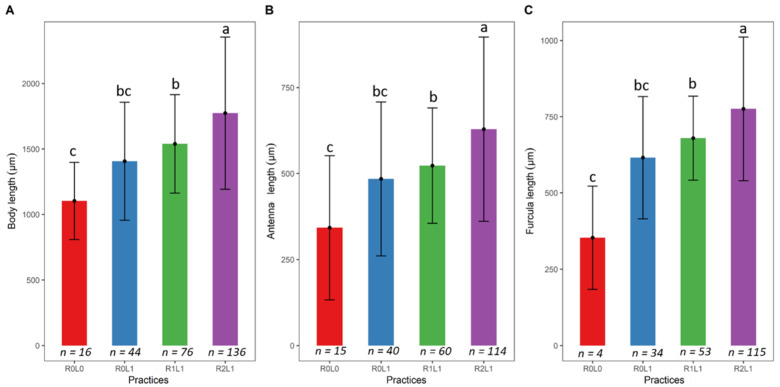
Average functional traits measured on the springtails according to the organic matter gradient. (**A**) Body length; (**B**) antenna length; (**C**) furca length. Different letters indicate significant differences according to the Tukey test. The vertical lines represent the standard error for each practice. R0L0—no residue or legume in the plot, R0L1—Legume (*Pueraria phaseoloides*) only, R1L1—Legume + stump + leaf + fine branches, R2L1–R1L1 + trunk.

**Figure 4 insects-13-00892-f004:**
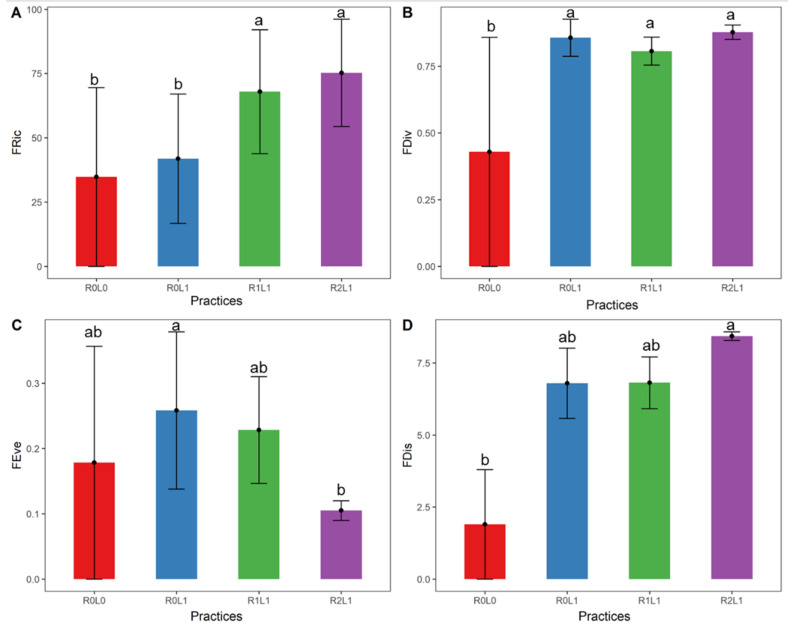
Average values of functional diversity indices of springtails according to practices. (**A**) FRic = functional richness; (**B**) FDiv = functional divergence; (**C**) FEve = functional evenness; (**D**) FDis = functional dispersion. Different letters indicate statistically significant differences between practices according to the Tukey test. The vertical lines represent the standard deviation for each practice (*n* = 3). R0L0—no residue or legume in the plot, R0L1—Legume (*Pueraria phaseoloides*) only, R1L1—Legume + stump + leaf + fine branches, R2L1—R1L1+ trunk.

**Figure 5 insects-13-00892-f005:**
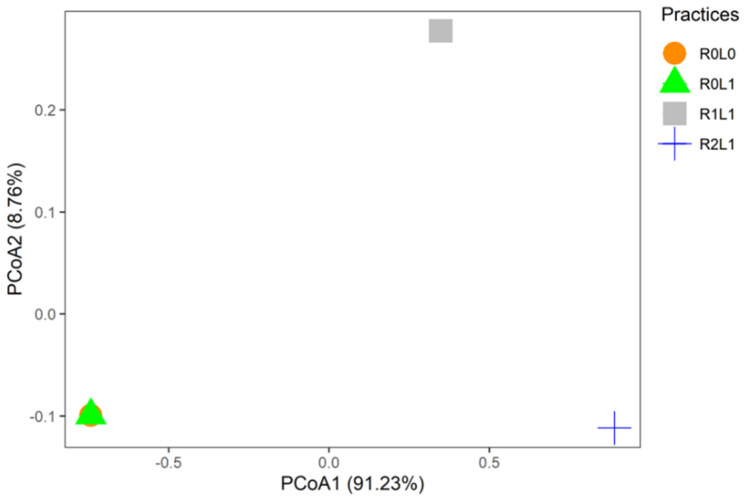
Principal coordinate analysis showing the patterns of functional composition of springtails communities in the different practices. This analysis is based on a Euclidean distance matrix. Points represent the centroid of replicates samples (*n* = 3). R0L0–no residue or legume in the plot, R0L1—Legume (*Pueraria phaseoloides*) only, R1L1—Legume + stump + leaf + fine branches, R2L1—R1L1+ trunk.

**Table 1 insects-13-00892-t001:** Description and attributes of morphological traits, considered for the trait-based approach of springtails communities.

Trait	Attribute
Body length	µm
Body modification	Body not modified
	Abdomen IV elongated
	Spherical body
Furca length	µm
Dens	Short
	Whip-shaped
	Long cylindric
Mucro	Very small straight
	Blade-like straight Bidentate Tridentate
Antenna length	µm
Ocelli	4 or 5 pairs of ocelli
	6, 7 or 8 pairs of ocelli
Post Antennal Organ	Absent
	Present
Pigmentation	Absent
	Diffuse
	Intense
	Pattern
Scales	Absent
	Present
Empodial appendage	Absent
	Present

## Data Availability

Due to privacy reasons, the data presented in this study are not publicly available. These data can be available on request from the corresponding author.

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
