# Peer review of "Using Trait-Based Approaches to Assess the Response of Epedaphic Collembola to Organic Matter Management Practices: A Case Study in a Rubber Plantation in South-Eastern Côte d’Ivoire"

_insects, 2022, doi:10.3390/insects13100892_

Round 1

Reviewer 1 Report

Lines 98-99

The geographic coordinates (5º 30′ 32.364” N 3º 32′ 51.755” W) are not in a cultivated area.

 Line 106

In Perron et al [12] the experimental design is not the same as the one used in the manuscript. In [12] it is said (on pg 3):

“Experimental plots were set up from December 2017 to January 2018. The experimental design was the same at SAPH and SOGB. It consisted of four treatments replicated four times in randomized blocks, giving 16 plots per site.”

The detailed experimental design must be described in this manuscript.

 Line 120

Sample collection must be accurately described:

How many samples were taken in each plot?

Is there no replication of the treatments?

 Line 126

The traps were left in activity only for two days? Why?

It seems like very little time to collect representative populations of springtails. For example, in the literature cited in the manuscript is said that pitfall traps were actives more days:

[39] Querner & Brucknet 2010 – pitfalls actives 14 days

[40] Driesen & Greenslade 2004 – pitfalls actives 7 days

 Line 157

Statistical Analysis – This paragraph indicates methodologies that do not correspond to statistical analysis, but are the methodologies used to measure the traits of the springtails. A new point of methodology should be made to explain how these traits were measured.

 Line 166-167

What statistics were used to check the normality and homogeneity of the variance of the data?

 Line 167-168

Explain what the following acronims means:

FRic, FEve, FDis, FDiv

 Line 196-197

It is not specified that an ANOVA has been done nor are its basic parameters indicated (sample size (n), Degrees of Freedon (df), F statistic).

 Line 200

It is not specified that an ANOVA has been done nor are its basic parameters indicated (n, df, F).

 It is not clear what data has been used in these analyses.

 Only three funtional traits are named  in this section. What about the other traits listed in Table 1?

 Line 219

Figure 2 – What is the sample size (n) in these graphs?

 Line 232

Figure 3 – What is the sample size (n) in these graphs?

 Line 237

The following variables are not defined in Metods:

functional richness

functional divergence

functional eveness

functional dispersion

Explain in Methods what the ecological meaning of these variables is.

 Line 259

Figure 4 – What is the sample size (n) in these graphs?

 Line 265

The variable “community weighted mean trait” is not defined in Methods.

What are the values of this variable in the samples?

 Line 289

Figure 5 – This PcoA is uninformative. Without replication of the tratments is not possible to say nothing statistically coherent.

 Line 329-332

Phrase too long and non-understandable. A separator is needed.

GENERAL COMMENTS

 The following aspects should be clarified:

 -The experimental design is not adequately explained.

 -It is not clear if the treatments have been replicated.

 -Pitfall traps have only been active for 2 days, which is too short a time to obtain representative data on springtail populations.

 -Statistical methodologies are not clearly explained.

 -The results of the statistical tests carried out are not indicated (ANOVAs: df, n, F statistic)

 -The graphs do not include basic data such as the sample size (n). Do the error bars represent the deviation of the treatment replicates? (it is not clear)

 -Table 1 indicates that 11 springtail traits have been measured, but there is only data for three of them. Basic data must be supplied, such as the average values of the different traits obtained in each plot or treatment.

 -The functional traits are not defined, nor their biological meaning in the context of the work carried out.

Reviewer 2 Report

The submitted manuscript presents a research focusing on the functional response of soil Collembola on organic matter input in an African rubber plantation system. The overall study is relatively straightforward and the data and conclusions are particularly valuable.

The study design and the sampling methods chosen are suitable, the statistical evaluation methods are also well chosen.

I consider it one of the important outcomes of the research, that the use of functional richness can be considered a good indicator if taxonomic determination encounters difficulties.  

Minor corrections:
Line 49 “mains” correctly “main”
Line 51 “functionning” correctly “functioning”

Reviewer 3 Report

A paper well done and with clear and explanatory figures.

In my opinion, it is a pity that they have not been able to contact a taxonomist to carry out parallel work with the species that appear.

My congratulations to the authors.

Reviewer 4 Report

Manuscript presented functional traits response of Collembola communities to various practice with logging residue on a  rubber plantation.  Although the presented study  are interesting, manuscript contain several  points  which should be clarified before accepted for publication.

The assumptions made in the work are missing.

i)                     First of all , there is  no defined the environmental gradient (= organic matter gradient). How was the content of organic matter assessed?  whether the examined gradient in organic matter was related to the management method used? How the amount of organic matter was measured, and how the quality, availability and quantity was evaluated?

ii)                   Examined functional trait response is limited only to epigeic Collembola, and not to whole Collembola communities, as in the title,  and it is a results of research method used with pitfall traps.

iii)                 In the material and methods chapter, the conducted experiment is too briefly presented, despite the reference to Perron's work, it makes it difficult to understand the research. For example, whether all studied plots were after the 40-year cycle of rubber plantation?  This information is only given in Conclusion. It is not clear if  the research was conducted only at sites after this cycle.

iv)                 Discussion on several points is unclear whether the authors conclude their own or others' results , i.e  line 312 – 313, and if their results are consistent or not with others, i.e 314-315.

detailed comments

Line 19-20 unclear, high functional diversity and large body size are related to which organic matter management practices?

Line 269 – 271 fragment is unclear, dissimilarity of functional composition between which plots? Why did you show distance among practices not among plots?

Line 363 – 364 – How was the functional composition pattern of epigeic Collembola related to ecological niches ?

Round 2

Reviewer 1 Report

Line 235: n = 18 ?

Lines 254, 257, 259: DF = 4 ?

Line 280: n = 3 ?

Line 310: n = 3 ?

If the experimental design was 16 plots and 6 traps on each plot (4 practices x 4 random blocks x 6 pitfall traps), I don’t understand why in some analyses n=18, or n=3, or DF = 4.

Lines 252 to 261: What is indicated in this paragraph does not agree with what is represented in Fig 4. Fdiv and Fdis seems to be changed in Fig 4.

For example, in this paragraph it is said that “Functional dispersion (FDis) was significantly different (ANOVA, F = 41.16; Df = 4; p = 5.97e-05) between practices with rubber residues and/or legumes (R0L1, R1L1 and R2L1) and those without (R0L0, Figure 4D).” but this is not true in Fig 4D.

It seems that what is indicated for Fdis in the text must be applied to Fdiv.

Lines 342-360: What is said in the discussion about Fdis and Fdiv is not consistent with what is said in the results.

Lines 368-371: What is indicated in the discussion about different Functional Composition between some practices is not consistent, since in the Results it is indicated that "These composition patterns do not seem to be affected by the effect of practices ..." (lines 288-289).

Lines 385-386: Inconsistent conclusion regarding results (lines 288-289).
